# Fuzzy Algorithmic Modeling of Economics and Innovation Process Dynamics Based on Preliminary Component Allocation by Singular Spectrum Analysis Method

**Alexey F. Rogachev *** , **Alexey B. Simonov** , **Natalia V. Ketko** and **Natalia N. Skiter**

Faculty of Economics and Management, Department of Information Systems in Economics, Volgograd State Technical University, 400005 Volgograd, Russia
* Correspondence: rafr@mail.ru

**Abstract:** In this article, the authors propose an algorithmic approach to building a model of the dynamics of economic and, in particular, innovation processes. The approach under consideration is based on a complex algorithm that includes (1) decomposition of the time series into components using singular spectrum analysis; (2) recognition of the optimal component model based on fuzzy rules, and (3) creation of statistical models of individual components with their combination. It is shown that this approach corresponds to the high uncertainty characteristic of the tasks of the dynamics of innovation processes. The proposed algorithm makes it possible to create effective models that can be used both for analysis and for predicting the future states of the processes under study. The advantage of this algorithm is the possibility to expand the base of rules and components used for modeling. This is an important condition for improving the algorithm and its applicability for solving a wide range of problems.

**Keywords:** singular spectrum analysis; fuzzy logic; innovative activity; time series; economic development; models





## 1. Introduction

An important task arising in many areas of economic science and practice is analysis and prediction of the dynamics of processes and phenomena. At the same time, there are many approaches to solving these problems, particularly those based on application of statistical methods and methods of data mining. Such methods as trend extraction (for example, based on correlation and regression analysis) and cyclical fluctuations (both based on indices and Fourier series decomposition), as well as the simplest methods based on use of average values, in combination, can be considered as methods of unsupervised learning. In the same group, we can include the majority of methods of data mining, primarily using different types of neural networks, as the simplest direct propagation networks, and usually more effective in analysis of the dynamics (recurrent neural networks, fuzzy neural networks, and others). A number of other methods, such as those simulating random walk or using smoothing, moving averages, the more complex ARIMA method, and LOESS methods, are based on construction of part of the model totality. Sometimes, these parts are highlighted in a preliminary analysis, as in the X-12 method. These approaches make it possible to identify the main regularities in the dynamics of most phenomena and make a qualitative forecast, at least for the short- and medium term.

However, in several sciences, particularly in economics, when studying the dynamics of some phenomena and processes, modeling becomes more complicated due to their characteristic chaotic behavior, a sharp change in trends, as well as the presence of a large number of different types of fluctuations that may vary over time. The reasons for such behavior may be the presence of many factors affecting the dynamics of the phenomenon under study and a complex, often cyclical nature of the relationship between these factors

as well as between them and the phenomenon under study. In such conditions, known methods do not have the necessary tools to build a predictive model with a high level of reliability of results. Indeed, methods based on analysis of the entire training totality (for example, correlation–regression analysis, Fisher's g test, fuzzy neural networks) can be ineffective in some cases because the trend and nature of fluctuations changed in the totality, which leads to low quality of obtained models. On the other hand, methods based on analysis of part of the training aggregate, such as the moving average or X-12, may be ineffective due to a lack of comprehensiveness of the obtained models.

In economics, such phenomena, which are characterized by close to chaotic behavior, often occur in stock exchange analysis, analysis of tourist flows, as well as in many types of long-term analysis, for example, in long-term analysis of electricity consumption or economic activity of enterprises [1]. The influence of uncertainty is especially great in modeling of innovation activity, which makes modeling in this area especially difficult [2,3]. Similar phenomena can be observed, for example, in forecasting of time series (TS) of crop yields, weather phenomena, availability of water resources, and so on [4].

When analyzing such phenomena, it seems effective to combine mathematical methods with methods of expert evaluation of existing trends (fundamental analysis), or to apply methods of intelligent data analysis. However, the results of applying expert methods not only depend significantly on the level of qualification of experts and their availability of the necessary information and tools but also lead to the risk of systematic errors, for example, in cases where experts are interested in this or that result of predictions. On the other hand, the prerequisites of the obtained results of intelligent analysis methods, such as different types of neural networks, are usually not transparent enough, which can lead to untraceable errors in determining the main trends and also not exclude system errors. For example, in the case when a bank gives a loan to an agricultural producer on the basis of the bank yield estimate made by artificial intelligence, it, in turn, affects the yield: the lower the bank estimate, the lower the amount of loans issued, which reduces the ability of agricultural producers to use technology to improve yields [4].

Based on the above reasoning, the need is obvious to improve the algorithms of studying the dynamics of economic phenomena using tools of modern methods of data analysis and statistical models. At the same time, statistical models, supplementing the possibilities of teaching methods with or without a teacher, will provide transparency of conclusions, their verification and validation, and facilitate application of created tools for analysis and forecasting not as "black boxes" but as understandable in a functioning mechanism. The described qualities are especially important in study of complex systems in which there may be elements of chaotic behavior. Sometimes, for such systems, understanding the logic of what is happening is no less important than the accuracy of the resulting forecast. However, such models are built on the basis of certain theoretical hypotheses and do not always allow to react quickly enough to changes in the trend, seasonal, and cyclical fluctuations to reveal hidden regularities in development of this particular phenomenon. In addition, the effectiveness of such models depends largely on the knowledge of the expert who creates them.

One of the tools that can be used to highlight such changes in trends and identify hidden components is the singular spectrum analysis (SSA) method [5–10]. This method allows us to decompose the original series of dynamics into basic components, which are a combination of a trend, cyclical fluctuations of different frequency, and random deviations. On the basis of the models built, it is possible to assess both the presence and properties of the main aspects of the dynamics and the relationship between the phenomena, as well as to build a forecast of the dynamics of the phenomena.

However, although the SSA method can be used to build interpolation and extrapolation models, its efficiency for solving some economic problems is not high enough. As shown in [5], the SSA method has lower efficiency than, for example, neural networks when solving problems of electricity consumption forecasting. In this connection, a number of authors suggest combining SSA with other methods (first of all, with methods of

intelligent data analysis) in order to improve the quality of extrapolation. The effectiveness of the combination technique is confirmed by the research in [5], where construction of a hybrid model in which the trend extracted by MSSA (multi-channel SSA) is processed by neural fuzzy network. Similarly, in [8], SSA was used to clean the training dataset from noise before training recurrent neural networks (RNN). When solving the problem of short-term net load forecasting, [9] proposes to use SSA as the method of preliminary dataset analysis, and the results obtained are extended with exogenous variables to obtain the best results of LSTM network usage. In some cases, such algorithms are less efficient compared to other hybrid algorithms; for example, when modeling the flow of tourists to Indonesia [10], the combination of SSA and fuzzy time series (FTS) generated worse results than the SARIMA-FTS combination. This can be explained by the fact that, in this study, SSA was used only to identify the trend and seasonal component while SARIMA also enables considering autoregression. From the point of view of considering the SSA method as a pre-processing tool, of interest is [11], in which the fuzzy C-means clustering (FCM) method was used before the SSA method, which enabled obtaining results in the FCM-SSA-ARIMA model that surpass the accuracy of SSA-ARIMA and ARIMA results.

Based on analysis of the effectiveness of various methods of studying the dynamics of economic processes, the authors in this article explored the possibility of using a combined algorithm for modeling and predicting TS in the economy and, in particular, in innovation, where the influence of uncertainty is particularly high. The proposed algorithm makes it possible to combine a transparent allocation of the main trends on the basis of search methods with an evaluation of the trends selected by expert systems to reduce the influence of human factors and to build models of dynamics based on the most significant selected trends (discarding unimportant elements of the dynamics series).

## 2. Materials and Methods

To solve the problems of forecasting complex economic and innovation processes in conditions of uncertainty, the authors propose to use a combined approach to build a multi-component model of their dynamics. Similar algorithms are used in radio engineering [12], which makes it possible to increase the efficiency of control and management processes in a competitive environment of functioning of adaptive information transmission systems, as well as to create support for the process of adaptive control of the state of radio devices.

As a similar process control tool for its application in the field of economics, we propose an algorithm, reflected in Figure 1.

The algorithm consists of the following steps:

1. Selection of time series components using the SSA method. For these calculations, we used the Pyts package [13].
2. Creation of fuzzy estimates of the possible character of time series components on the basis of search analysis tools and empirical rules. This estimation takes place in the module "phasifier" and can be performed both in automatic mode and corrected by expert method taking into account the proposed automatic estimations. These calculations were performed using packages NumPy, ScyPy, and Pandas. The package Matplotlib was used for visualization.
3. The dynamic expert system, based on fuzzy logic, draws conclusions about the optimal classification for each component of the time series. At the same stage, the expert system can draw conclusions from the subject area, concerning, for example, the possible economic content of each component. In addition to the previously mentioned tools, we used FisPro 3.8 [14] to visualize and verify the calculations.
4. Expert system evaluations can be made in the form of fuzzy logic, in which case they must go to the defuzzifier to obtain clear evaluations.
5. Based on clear estimates, the optimal model of each component is selected and constructed, taking into account its type and possible economic content.
6. A generalized model, which can be used for analysis and forecasting of economic and innovative processes, is constructed.

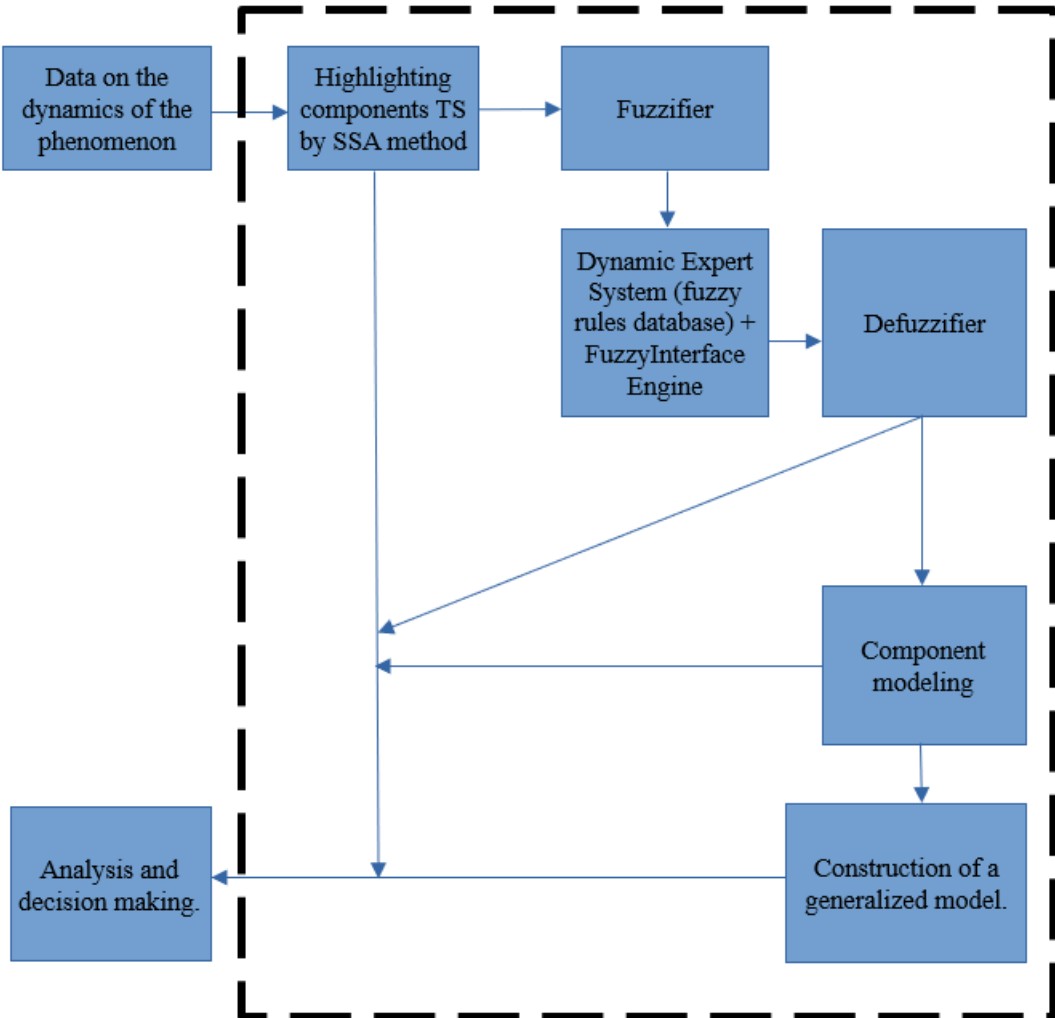

**Figure 1.** A generalized structural diagram of the dynamics analysis based on fuzzy processing of time series components obtained with SSA.

Let us consider the above steps in more detail.

Separation of the dynamics components allows us to consider SSA as a method of exploratory analysis, application of which is highly effective in cases where a dynamics series may include a large number of different components with insufficiently accurately studied properties. For example, as shown in [7], in many time series describing the dynamics of innovation activity in the Russian Federation and Volgograd region for twenty years, we can distinguish a trend, two cyclic components (with the period of about 3 years and about 8 years), as well as various types of autocorrelation and, of course, "white noise". This agrees with the general theoretical ideas about the structure of series of dynamics in the economy, in which analysis of the dynamics by years traditionally distinguishes short-term cycles of Kitchin (with a period of 2–4 years); medium-term economic cycles of Juglar with a period of 7–11 years and long-term cycles of Kuznets (period 15–25 years) and Kondratiev (period 45–60 years), while long-term cycles in our study [7] were not distinguished because of the limited length of time series.

Before we start to use SSA method, we should set window's length L [15]. Large values of L allow reconstructing long period oscillations, but too large values of L leave very few observations to make qualitative reconstruction. Therefore, L usually takes a value from quarter to half the time series length (N/4 . . . N/2) [16]. We can also examine the median value [17] as the optimal L value.

The SSA method consists of two phases, decomposition and reconstruction [18]. Let us consider these stages of the method.

The first stage of decomposition phase is embedding. The original time series $F = (f_1, f_2, \ldots, f_N)$ of length N is transformed into a sequence of multidimensional vectors (trajectory matrix X) during this stage. The matrix X has the dimension L × K, where K is number of the lagged vectors selected:

$$K = N - L + 1 \tag{1}$$

Therefore, K is the number of columns and L is the number of rows of the trajectory matrix X. Each column of this matrix is rolling window of length L, so 1st column has elements of the time series F from the first till L, 2nd column has elements from the second till L + 1, and so on.

$$X = \begin{pmatrix} f_1 & \cdots & f_{N-L+1} \\ \vdots & \ddots & \vdots \\ f_L & \cdots & f_N \end{pmatrix} \tag{2}$$

X is matrix with equal elements on all diagonals (so it may be considered as a Hankel matrix if $L = N - L + 1$).

The second stage of the decomposition phase is singular value decomposition (SVD) of trajectory matrix. The result of this stage is the decomposition of the trajectory matrix X into elementary parts, i.e., representation of the matrix X as a sum of elementary matrices. Let $\lambda_1, \lambda_2, \ldots, \lambda_L$ denote the eigenvalues of the matrix $S = X \times X^T$ taken in non-decreasing order, and d is number of $\lambda_i > 0$. Let us also denote $U_1, U_2, \ldots, U_L$ the eigenvectors of the matrix S, corresponding to the ordered eigenvalues [19].

Then, the singular value decomposition for X is

$$X = \Sigma X_i \tag{3}$$

$X_i$ is the elementary matrix with rank 1. These elementary matrices are calculated by the formula:

$$X_i = \sqrt{\lambda_i} U_i V_i^T, \tag{4}$$

where elementary vectors $V_i$ can be found after we calculate U and $\lambda_i$, using the formula:

$$V_i = X^T \frac{U_i}{\sqrt{\lambda_i}} \tag{5}$$

After decomposition, we have reconstruction phase. To reconstruct time series, we use diagonal averaging. Each matrix $X_i$ we obtained after decomposition phase is translated into a series of length N. We suppose L* = min (L, K), K* = max (L, K). For each element x of matrix $X_i$, let us also denote x*$_{ab}$ = x$_{ab}$, if L < K, and x*$_{ab}$ = x$_{ba}$, if L > K. Diagonal averaging [18] transforms each resulting matrix Xi into a series according to the formula for each element $\widetilde{f_{ik}}$ of vector $\widetilde{F_i}$:

$$\widetilde{f_{ik}} = \begin{cases} \frac{1}{k+1} \sum_{n=1}^{k+1} x^*_{n,k-n+2}, & \forall\, 0 \le k < L^* - 1 \\ \frac{1}{L^*} \sum_{n=1}^{L^*} x^*_{n,k-n+2}, & \forall\, L^* - 1 \le k < K^* \\ \frac{1}{N-k} \sum_{n=k-K^*+2}^{N-K^*+1} x^*_{n,k-n+2}, & \forall\, L^* - 1 \le k < K^* \end{cases} \tag{6}$$

This formula averages the elements along the diagonals. Therefore, applying diagonal averaging to the resulting matrices $X_i$, we obtain the series $\widetilde{F_i}$. These series can be considered as additive components of the time series f that was under study. These components may be classified as a trend, cycle, or noise component [17], and then we can model it.

In spite of its significant advantages, the SSA method requires fairly good expert analysis of the components of a time series in order to correctly assess their content. This is a rather labor-intensive process due to the large amount of work on expert evaluation in the comparative analysis of the dynamics of a large number of phenomena and processes,

presumably related to each other (which is possible and can lead both to construction of quantitative models and to identification of fundamental features of the processes under study, as we showed in [7]). Expert assessments are also complicated by possible changes in trends, changes in the nature of fluctuations, as well as the complexity of expert separation based on visual assessment of short-term cyclical fluctuations and autocorrelation. In addition, as analysis of the costs of innovation activity in Volgograd region showed, SSA does not always effectively separate the weakly expressed trend and medium-term (long-term) fluctuations, as a result of which one component must be divided into two mathematical models.

To facilitate and partially automate the expert evaluation in our algorithm, we propose to use the toolkit of fuzzy sets, which will be implemented in the phasifier and the database of fuzzy rules. This toolkit, based on fuzzy inference rules, will allow you to determine the dynamics model for a particular component (or propose several of the most appropriate models for experts to choose from), as well as to make preliminary conclusions about the possible economic content of specific components on the basis of connected databases and/or knowledge bases.

It should be noted that, in [6], a tool for identifying the types of dynamics (trend and cyclic component) on the basis of modeling is proposed for similar problems; however, this approach does not allow expanding the set of rules and including, for example, rules for trend changes or rules for analysis of cyclic fluctuations, which are not sinusoidal. Another approach, effective from the point of view of extracting the cyclic component, is a combination of SSA (MSSA) and Fisher filter methods [20]. For example, [21] suggests using SSA as a preprocessing method to increase the efficiency of detecting hidden periodicities in the phenomenon under study. On the other hand, in their paper [18], the authors use logic close to that proposed by the authors in the algorithm, according to which "Fisher g test <is used> to select the principal components to be aggregated in the reconstruction step". However, even in this approach, there are difficulties with the unclear economic content of individual economic cycles reconstructed on the basis of Fisher's g test, as well as with the impossibility of changing the model used and combining, for example, several approaches to model the cyclicality of different components.

Based on this review, we can conclude that the task of finding a way to extend the toolkit for recognition of SSA series components is relevant, and the toolkit for phasing and applying fuzzy inference rules, which is easily extensible and customizable, is presumably effective enough to solve this problem.

The fuzzifier converts the indicators extracted by SSA (in particular, the values of the most significant criteria) into fuzzy linguistic variables belonging to fuzzy sets Aj (individual for each indicator). As a rule, these variables have an affiliation function (AF), for example, of the triangular form

$$\mu_A(s) = \begin{cases} 1 - \frac{|s-c|}{d} & \forall s \in [c-d, c+d] \\ 0 & \forall s \notin [c-d, c+d] \end{cases}, \tag{7}$$

where $c$ is the center point of the triangle; $d$ is the width of the triangle.

The peculiarity of this transformation is that we can construct fuzzy estimates for any criterion, and, if necessary, make fuzzy estimates based on expert reasoning.

After the transformation, formation of fuzzy conclusions about the possible type of dynamics of the phenomenon defined by the SSA component under study takes place. For this purpose, additive convolution is used, which is performed on the basis of the rules of the form:

$$\text{if S1 is A11 and S2 is A21, then Z is B,} \tag{8}$$

where S1 is a fuzzy value for one indicator, S2 for another indicator, and so on; Z is a fuzzy value for the resulting component type.

For example, "if Spearman criterion value is high and L value obtained by Foster-Stewart method is high, then presence of trend with positive first derivative is very high".

The set of rules contained in the database of fuzzy rules and/or knowledge base, if necessary, can be adjusted, which is an important distinguishing feature of this algorithm, giving it high flexibility.

Based on the obtained fuzzy inferences, the defuzzifier transforms the fuzzy set of obtained linguistic evaluations for the inferences into a ranked list of possible trend types for a given component.

On the basis of the obtained estimations, the method of modeling of this component can be automatically chosen. In particular, we used statistical models of trend, cyclical fluctuations, and autocorrelation of deviations. Construction of SARIMA models also proved to be effective.

To put it briefly, the authors proposed an algorithm, shown in Figure 1, as a sequence of known methods. Technical work on the information system design has also been carried out. We also developed a fuzzy rules database for an expert system and tested the results of applying the proposed algorithm.

From the authors' point of view, the main advantage of this algorithm is that we can exclude specific values of a random component from the final model, which will reduce the effect of overtraining and enable better assessment of model error.

It is also necessary to mention the fact that, alternatively, different kinds of neural networks can be used at the modeling stage. In this case, since the data in the components, as a rule, are cleared of random deviations and contain only one trend, training on such datasets will be highly effective [8].

The peculiarity of the presented algorithm is the possibility to offer economic estimates depending on which components are present in the modeled time series. Thus, the absence of cycles with a period of 7–11 years (Juglar cycles) may indicate difficulties with effective replacement of production capacity.

## 3. Results

The algorithm developed by the authors was initially tested on a generated dataset consisting of a polynomial trend, three harmonics, and a random component with negative autocorrelation with lag 1. The SSA method efficiently extracted all the main components, which allowed us to identify the types of dynamics with a high level of confidence and build a model that described with high accuracy the functions used to generate the dataset.

After testing, the authors used the developed algorithm to analyze information about GDP and the number of researchers in the Russian Federation. The information for the analysis was taken from the website of the Federal State Statistics Service of the Russian Federation [22].

To build the model of GDP dynamics of the Russian Federation, the data for 1995–2020 were used; the data for 2021 were used as a test dataset, on the basis of which the quality of the forecast built by the created model was determined. To eliminate the effect of inflation, the data were adjusted to 2020 prices.

The results of modeling were compared with several statistical methods, including trend-seasonal models SARIMA and ARIMA. Of these methods, ARIMA (0,2,0) provided the best results, and we used them as a basis for comparison with the developed algorithm results. The comparison with the methods of intellectual analysis, in our opinion, is not entirely correct since the final result of the developed algorithm is to obtain statistical models with a specific economic interpretation and not just the forecast itself.

As a result of the work of the phasifier, it was found that it does not recognize fluctuations whose period is comparable with the length of the studied series of dynamics well enough. In our case, one of the highlighted SSA components was a cycle with a period of about 20 years (the Kuznets cycle), with a total length of the studied period of 26 years, and this cycle was not effectively recognized until we changed the rules to a set of estimates of the presence of the cycle according to data, first and second final difference, and periodogram. After introducing additional rules, the algorithm correctly recognized

this long cycle (see Figure 2a). Then, a cycle model with a period of 22 years based on the sine function was constructed in automatic mode.

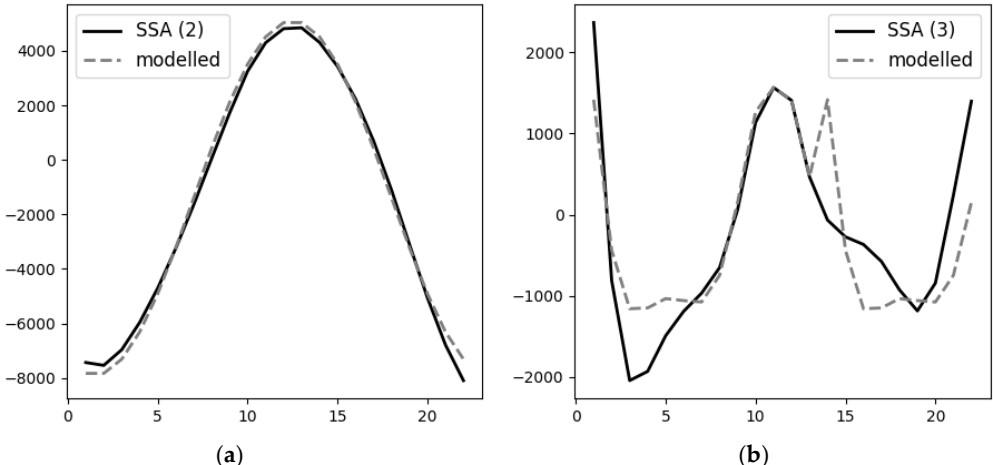

**Figure 2.** Modeling results for the 2nd (**a**) and 3rd (**b**) components extracted by the SSA method.

Difficulties arose in study of the third component, recognized as random deviations with positive autocorrelation, but they were expertly estimated as cyclical fluctuations with a period of about 10 years (Juglar cycles). These cycles have a rather gentle growth and a sharp decline, which complicates their identification, recognition, and modeling. As a result, for effective recognition of such cycles, it was proposed to use in the phasifier an analogue of the Henderson filter to the selected components, which reduced the influence of possibly inaccurately defined values at the beginning and end of the components but does not significantly affect the overall result of recognition. Then, a rule based on detecting the cyclic direction of function change was added. This allowed us to identify this type of cycle and model it by index method (although it was possible to use other methods as well), which provided acceptable accuracy results of component recognition (see Figure 2b, unexplained variance fraction 48.2%). Since insufficient data do not allow modeling seasonal components by analogy with X-12, an alternative could be use of neural networks, but the possible quality of training on such a small dataset raises doubts about the reliability of the obtained result. The classical index method used in the study, on the other hand, is fairly easy to automate and reduces error associated with the decadal cyclic component.

To evaluate the effectiveness of the algorithm developed in this study, we compared the created model with some statistical models, including models with polynomial trend, SARIMA and ARIMA models, and Fourier series decomposition (with maximum six series). The ARIMA model showed the best results, so we will compare them with the results obtained by the presented algorithm. As can be seen from Figure 3, the model created with the help of the presented algorithm interpolates quite accurately the value of RF GDP in the middle of the series, deviating significantly from the real values in the periods of crises (which is generally characteristic of statistical models). Even in these periods, in general, the model proved to be more efficient than ARIMA. At the same time, both models have high enough accuracy; if we evaluate them on the basis of the proportion of the explained variance $R^2$, it is estimated as 0.973 for ARIMA and 0.985 for the model created before the change in fuzzy recognition rules. After creating rules that account for the variation in the amplitude of the oscillations and the complex nature of the Juglar cycle, the $R^2$ estimate rose to 0.992. Moreover, even the model of the first three components provided an $R^2$ value of 0.976, which shows interpolation better than the ARIMA method. The forecast error for 2021 for the created model was +2.2% compared to −3.4% for ARIMA, but the accuracy of this forecast is not high also because of the COVID-19 crisis (as mentioned above, crisis phenomena reduce the accuracy of the model).

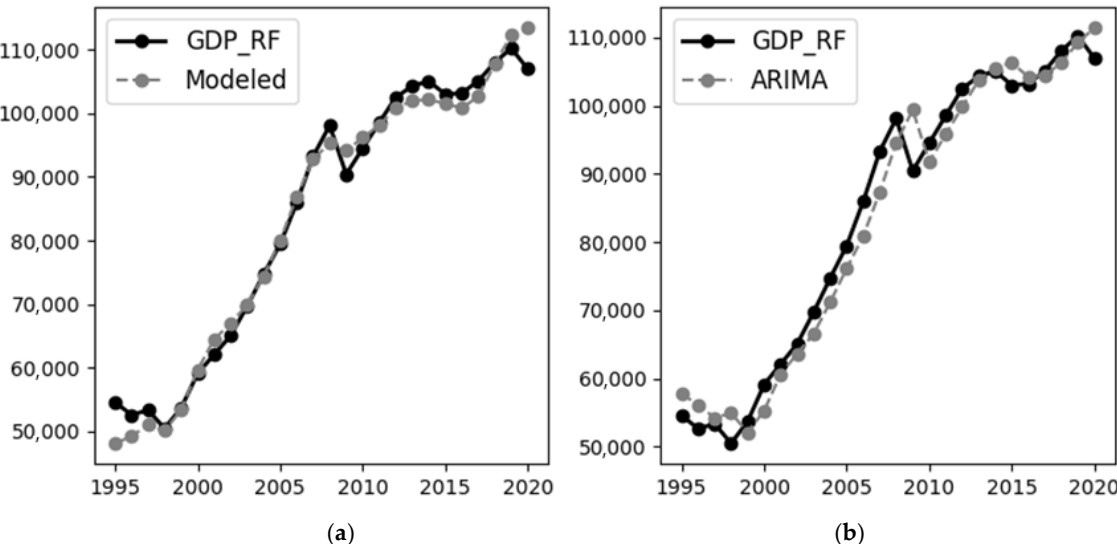

**Figure 3.** Results of the comparison of the model of GDP dynamics of the Russian Federation (in constant prices), developed with the algorithm (**a**), and ARIMA (**b**).

We also compared the results of our algorithm with the results of the SSA + LSTM algorithm, which was studied in [8,9]. In those cases, SSA was also used as the method of preliminary dataset analysis and the result obtained was used to train the LSTM network. We used TensorFlow [23] as an end-to-end open-source platform for machine learning and scikit-learn [24] to build and use a network. The result of this algorithm was worse both than ARIMA and the method proposed. SSA + LSTM had an $R^2$ value of 0.960, but, in our case, we had a small dataset, so the LSTM network lacked enough data to train. Therefore, we cannot conclude that the SSA + LSTM algorithm is worse than ours, and the effectiveness of them is not constant but depends on inner components. Therefore, the SSA + LTSM effectiveness depends on hidden element number and input window size, and our algorithm effectiveness highly depends on fuzzy rules database content. Hence, both algorithms can progress to improve their effectiveness, so we cannot affirm which is the best.

When analyzing the dynamics of the number of researchers in the Russian Federation (you can see the data at Supplementary Materials, Table_S1.csv and Table_S2.csv), we also obtain six components of time series using SSA (Table_S3.csv). After modeling them (Table_S4.csv, Table_S5.csv, Table_S6.csv, Table_S7.csv, Table_S8.csv, Table_S9.csv) we can assert, that the accuracy of the model (Table_S10.csv) was much lower than in the analysis of GDP. This is quite expected, since the number of researchers refers to indicators of innovation activity, which is characterized by high quantitative and qualitative uncertainty.

Obvious fuzzy rules quite effectively distinguished cyclic components and assumed the presence of a quadratic trend. Difficulties appeared in analysis of the second component, where a sharp change in the trend is modeled (from growth in the number of researchers in 2000–2012 to a sharp decrease in 2012–2020). The proposed rules based on the difference in moving averages with different periods showed a possible trend change, but the algorithm highlighted as more preferable the autocorrelation model with lag 1. Theoretically, this could have been changed by adjusting the rules to reduce the preference of the autocorrelation model as a whole, but this would have required a readjustment of the entire algorithm, so a rule tracking a single change in the trend was introduced. This situation confirmed that the question of the optimal approach to trend change tracking using the proposed algorithm remains open.

Note that both the autocorrelation model and the trend change model were quite effective in interpolation (see Figure 4). The model constructed using the original rules, which does not directly incorporate the trend change and indirectly estimates it through autocorrelation, has $R^2 = 0.810$ fractions of the explained variance. This is higher than a

similar estimate of 0.726 for the ARIMA (0;2;0) model considered, and the ARIMA model shows the best results among the statistical methods we applied.

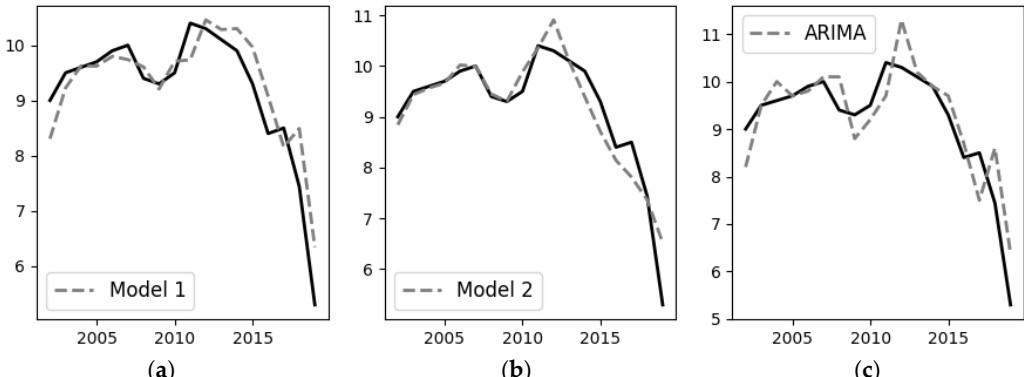

**Figure 4.** Results of comparison of the model of innovation activity in Russia developed with (**a**) the algorithm without the rule for trend change (Model 1), (**b**) with trend change (Model 2), and (**c**) ARIMA.

However, the prediction accuracy of both models turned out to be low: they significantly underestimated the share of innovatively active organizations in 2020 (the created model predicted only 2.8% of innovatively active organizations estimated according to the 3rd Oslo Manual methodology, and ARIMA 3.15%), which stabilized and even slightly increased (estimated according to the 3rd Oslo Manual methodology to 6.29% from 5.30% in 2019). This once again proves the low quality of predictions of models of innovation during bifurcations.

At the same time, the model built with the trend change in mind was not only more accurate in interpolation ($R^2$ = 0.875) but also less pessimistic in estimating the forecast for the next year (at 5.59%). In other words, the trend change model was obviously more effective in making forecasts and from the analytical point of view.

As well as for the previsions dataset, we also compared the results of our algorithm with the results of the SSA + LSTM algorithm [8,9]. This time, the dataset was even worse for LSTM network training. We had fewer data and we had many cycles with long periods that are hard for network estimation. Therefore, the result of the SSA + LTSM algorithm that we obtained by searching the best settings was an $R^2$ value of 0.198. This result does not indicate low effectiveness of the SSA + LSTM algorithm. This algorithm is simply not efficient enough to study data similar to the ones we are studying in this case.

The results of calculating the quality of the models are shown in Table 1.

**Table 1.** The results of the model comparison based on $R^2$.

| | ARIMA (0,2,0) As the Best of Statistic Models of the Analyzed Data | Algorithm with the Base Rules | Algorithm with the Modified Rules | SSA + LTSM |
|---|---|---|---|---|
| GDP dynamics in Russia | 0.973 | 0.985 | 0.992 | 0.960 |
| Innovation activity in Russia | 0.810 | 0.726 | 0.875 | 0.198 |

The algorithm created made it possible not only to create a model that interpolates the available data accurately enough and made it possible to create a relatively accurate forecast; it also highlighted the trend and the three cyclical components. This has obvious analytical value, particularly making it possible to relate the highlighted cycles to various

economic processes and assess the peculiarities of the development of these processes by studying the behavior of these components.

## 4. Discussion

The proposed algorithm enables us to build a fairly accurate model of the dynamics of economic phenomena, which can be used both for analytical purposes and to build short-term forecasts. Unlike analogues, the main advantage of this algorithm is the possibility to automate it efficiently, as well as to extend it by adding new rules (in the phasifier) and new models (in the dephasifier). Theoretically, this may allow the user who is not fully familiar with the tools of statistical analysis and data analysis to build highly accurate models. However, as shown in the study, this requires an extended rule base, including tools for analyzing periodic fluctuations that are not sinusoidal. It also makes sense to supplement the base with rules that allow to determine the change in the trend, the appearance and disappearance of periodic fluctuations (for example, in our study, the amplitude of short-term fluctuations has been sharply decreasing in the last ten years, which requires an appropriate economic explanation). Perhaps integration of a knowledge base into the algorithm, which will be able to identify the most typical economic causes of certain types of dynamics components, will make it possible to assess the change in these components.

One of the directions of research that seems promising is extension of this algorithm to apply it with the MSSA method, which can be effective in solving a number of economic problems. On the one hand, the super-complexity of economic systems leads to the fact that inclusion of additional factors does not always improve the quality of the economic model, but, for a number of problems (where the condition is sufficiently cleared of cross-correlation, in the presence of lag variables, in real-time estimation), use of MSSA can be quite effective.

Another direction for further improvement of the algorithm is to build special rules to identify and simulate crises. The expected manner to realize further improvement of the algorithm may be diagnosis of significant deviations in real values from those predicted by the model (as has been shown, these deviations tend to grow rapidly during a crisis) and construction of typical scenarios of crisis development and bifurcations on the basis of historical data. Here too, application of the MSSA method to solve the above questions with further automatic recognition of the obtained results seems promising.

Moreover, an interesting area of research is application of the considered algorithm for "nowcasting", predictions based on real-time data, similar to [18]. As shown, for example, in works on real-time simulation of the output gap [18,25], the task of real-time simulation is associated with considerable uncertainty; however, its results are highly demanded in the decision-making process.

A special issue is the need to assess the possibility of retraining the model while applying the proposed algorithm. On the one hand, two to four parameters are used in modeling each of the components, resulting in a very large total number of parameters. On the other hand, each of the models, in fact, is not a model of the original series but a model of some of its components. Thus, each model contains random deviations and does not deprive the original data of degrees of freedom, so it makes sense to estimate the degrees of freedom of each component separately, in which case we can argue that model overfitting does not occur. Moreover, even the models of the first three components allow achieving sufficiently high accuracy, and adding additional components to the model makes sense from the point of view of economic analysis rather than ensuring interpolation accuracy.

In general, we can conclude that the algorithm proposed by the authors demonstrated sufficiently high efficiency in solving a specific economic problem, but, for practical applications in various areas of the economy, it requires further development and additional adjustment, on the one hand, in creation of new rules of recognition of economic trends, often taking into account the specific subject area, and, on the other hand, towards studying the best options for preprocessing data and creating effective models (paired with the phasing tools used). The main advantage of this algorithm is that it can be easily modified

due to the modular structure and flexibility of fuzzy sets tools used in this algorithm. This is especially important in the current context of striving for sustainable development, including "innovation resilience behavior", which implies ensuring the stability and adaptability of both economic systems and the methodologies used in their management [26]. The created algorithm was also tested for the tasks of modeling the GDP of the Russian Federation and the number of researchers in the Russian Federation, as a result of which qualitative models of the main economic cycles were built and a complex model that allows describing the available data and making a forecast for the next year was created. The accuracy of interpolation and extrapolation was higher than, for example, when using the ARIMA method.

This algorithm can be used in studying the dynamics of complex systems, primarily in economics, particularly in study of innovative processes.

## 5. Conclusions

Study of application of the developed algorithm based on fuzzy logic methods and subsequent statistical modeling for post-processing of the components extracted by SSA showed that the proposed approach is quite effective when working with the generated data. However, when solving the economic task of modeling and forecasting the GDP of the Russian Federation, it was found that more complex rules are required when evaluating the components extracted by SSA, as well as use of special preprocessing of these components. The algorithm resulted in an effective descriptive model, which can be used both for analytical purposes and for forecasting purposes. However, to increase the efficiency of this algorithm in solving various economic problems, its further development, tuning, and modifications are required.

The models created on the basis of application of the proposed algorithm confirmed that innovation processes are associated with a higher level of qualitative uncertainty compared to other economic processes. The models built to describe the dynamics of a number of researchers are more complex and require addition of fuzzy component recognition rules. Moreover, both the constructed models and ARIMA models (as well as other models of dynamics that were used for comparison) turned out to be much less accurate than when forecasting the dynamics of GDP of the Russian Federation. All this proves the high degree of complexity of the tasks of analysis of innovation activity. Nevertheless, the proposed approach of combining SSA and fuzzy estimation has coped quite effectively with this difficult task as well.

Although the SSA method is widely used for solving problems in many fields, its application in economics, and, especially, innovativeness, is not wide enough. In our opinion, this is due to considerable complexity of economic processes and regularities, which increases the role of the expert in application of SSA (as well as most other mathematical methods) in economics. However, use of an expert, particularly for recognition of the results of the SSA method, raises a number of problems related to its qualification, subjectivity, impartiality, and so on. We hope that the above algorithm, at the expense of the built-in base of fuzzy rules and possible explanation of the economic content of the selected components, which includes some kind of expert system, will help to solve this problem and enable wider application of the promising SSA method in further economic research.

## 6. Patents

Simonov A.B., Rogachev A.F. Certificate of registration of a computer program RU 2021619629. Analysis of dynamic indicators of socio-economic systems by the SSA method. Registered on 15 June 2021. Application No. 2021618587 2 June 2021.

**Supplementary Materials:** The following supporting information can be downloaded at: https://www.mdpi.com/article/10.3390/a16010039/s1, Table_S1.csv: Dynamics of innovation activity in Russian Federation (%), data; Table_S2.csv: Dynamics of innovation activity in Russian Federation (%), data, prepared for SSA analyze; Table_S3.csv: Innovation activity in Russian Federation (%), components of time series, obtained throw SSA analyze; Table_S4.csv, Table_S5.csv, Table_S6.csv,

Table_S7.csv, Table_S8.csv, Table_S9.csv: Results of components (C1–C6) of innovation activity dynamics in Russian Federation (%) recognition and modeling; Table_S10: Comparison of models of innovation activity dynamics in the Russian Federation (%).

**Author Contributions:** Conceptualization, A.F.R. and A.B.S.; methodology, A.F.R. and A.B.S.; data curation, N.V.K., N.N.S. and A.B.S.; writing, A.F.R., N.V.K., N.N.S. and A.B.S.; supervision, A.F.R., N.V.K. and N.N.S. All authors have read and agreed to the published version of the manuscript.

**Funding:** This research received no external funding.

**Data Availability Statement:** Data have been collected from https://rosstat.gov.ru/ (accessed on 1 December 2022).

**Conflicts of Interest:** The authors declare no conflict of interest.

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
