# Peer review of "Fuzzy Algorithmic Modeling of Economics and Innovation Process Dynamics Based on Preliminary Component Allocation by Singular Spectrum Analysis Method"

_algorithms, doi:10.3390/a16010039_

Round 1
Reviewer 1 Report
This paper proposed an algorithm for analyzing and forecasting time series of economics and innovation processes.
The proposed algorithm is a hybrid between a singular spectrum analysis method and a fuzzy expert system.
Their approach is quite reasonable because the fuzzy expert system can reduce the use of some experts to recognize the results of the SSA part.
(1) The abbreviation SSA in the title should be replaced by the complete spelling.
(2) Section 2 includes many things. We can not distinguish the new idea proposed in the paper.
Is it possible to clarify which points are new?
(3) Section 3 (Results) explained the accuracy, for example, R^2, in the text. The authors should summarize the result in a table for a more understandable comparison.
(4) l.194 Hankel matrix is a square matrix. So, X is not a Hankel?
Author Response
The authors are grateful to the reviewer for the constructive comments sent.
Response to Reviewer Comments
Point 1: The abbreviation SSA in the title should be replaced by the complete spelling.
Response 1: The name has been changed (supplemented). The meaning of the abbreviation SSA is described and the abbreviation itself is introduced in line 97 (new)
Point 2: Section 2 includes many things. We can not distinguish the new idea proposed in the paper. Is it possible to clarify which points are new?
Response 2: At the end of Section 2, we added a short summary (lines 292-295). It reflects that the authors proposed a new algorithm, shown in Figure 1, as a sequence of application of the methods under consideration. Technical work has also been carried out on designing an information system, creating a database of rules for an expert system, testing the results of applying the proposed algorithm.
Point 3: Section 3 (Results) explained the accuracy, for example, R^2, in the text. The authors should summarize the result in a table for a more understandable comparison.
Response 3: The simulation results are reflected in the added table 1 (line 413).
Point 4: l.194 Hankel matrix is a square matrix. So, X is not a Hankel?
Response 4: Thanks for the comment. It was written that if X is a square matrix, it is a Hankel matrix
Reviewer 2 Report
The primary concern about this manuscript is the application shown in Section 3 (results). We know that the ARIMA model is stochastically linear and is not supposed to describe more complex patterns, as discussed in Section 1. Besides, the ARIMA(0,2,0) model only describes white noise after removing some local quadratic trends. Hence, the authors must compare the suggested method with all the competing models cited in Section 1. The authors should consider in this study more complex time series data.
Some minor comments:
(i) The meaning of the acronym SSA is missing;
(ii) The notation for the matrix X in equation 2 does not agree with the rest of the text.
(iii) The text does not make it clear how the V_i vectors relate to the equation (4)
(iv) What does the "*" operation mean?
(v) On Lines 211 and 212, using the superscript * conflicts with the "*" operator.
Author Response
The authors are grateful to the reviewer for the constructive comments sent.
Response to Reviewer Comments
As part of the response to the comments, we noted (line 322) that various statistical methods were used for comparison, and ARIMA (0,2,0) gave the best results in modeling. Comparison with the methods of intellectual analysis reflected in section 1, in our opinion, is not entirely correct, since the final result of the developed algorithm is to obtain statistical models with a specific economic interpretation, and not just the forecast itself.
Point (i) The meaning of the acronym SSA is missing;
Response 1: A transcript of the abbreviation SPA has been added to the title of the article and the abbreviation itself has been introduced in line 97 (new)
Point (ii) The notation for the matrix X in equation 2 does not agree with the rest of the text.
Response 2: The authors have tried to correct the remark in the corrected text of the article.
Point (iii) The text does not make it clear how the V_i vectors relate to the equation (4)
Response 3: Thanks for the comment. The index i was omitted in the equation.
Point (iv) What does the "*" operation mean?
Response 4: Thanks for the comment. "*" was incorrectly used as a designation for matrix multiplication (matrix product). It was changed in the corrected version of the article .
Point (v) On Lines 211 and 212, using the superscript * conflicts with the "*" operator.
Response 5: After eliminating the previous remark, the conflict should not arise.
Reviewer 3 Report
In the reviewed article, a modified algorithm for analyzing and forecasting TS economy and innovation processes is proposed.
The authors propose an algorithm combining the methods of singular spectral analysis (SSA) and fuzzy expert system.
The approach proposed by the authors is justified, since an expert system based on fuzzy logical inference can be useful for recognizing the results of SSA.
Some minor comments:
1. The abbreviation SSA in the title and the first mention in the text should be given in full
2. The extensive presentation of Section 2 in the article complicates the understanding of the novelty of the approach proposed by the authors.
3. It is necessary to explain the content of the operation "*".
4. It is necessary to formulate the areas of preferential use of the TS analysis and forecasting method proposed by the authors.
Author Response
The authors are grateful to the reviewer for the constructive comments sent.
Response to Reviewer Comments
Point 1: The abbreviation SSA in the title and the first mention in the text should be given in full
Response 1: The name has been changed. The meaning of the abbreviation SSA is described and the abbreviation itself is introduced in line 97
Point 2: The extensive presentation of Section 2 in the article complicates the understanding of the novelty of the approach proposed by the authors.
Response 2: At the end of Section 2, we added a local summary (lines 292-295). It reflects that the authors proposed an algorithm, shown in Figure 1, as a sequence of application of the methods under consideration. Technical work was also carried out on the design of the information system, the creation of a database of rules for the expert system, testing the results of the application of the proposed algorithm.
Point 3: It is necessary to explain the content of the operation "*".
Response 3: Thanks for the comment. "*" was incorrectly used as a designation for matrix multiplication (matrix product). It has been fixed.
Point 4: It is necessary to formulate the areas of preferential use of the TS analysis and forecasting method proposed by the authors.
Response 4: As part of this remark, we once again reflected at the end of Section 4 (line 490) that this algorithm, in our opinion, can be used in studying the dynamics of complex systems, primarily in economics, in particular in the study of innovative processes.
Round 2
Reviewer 1 Report
I confirmed the improvement in the revised version.
Author Response
The authors thank the reviewer for the useful interaction in improving the article.
Reviewer 2 Report
The model with the best fit is sometimes adequate. However, the point is that the authors compare their suggested approach with a simple linear model (ARIMA(0,2,0) is just twice-integrated noise). The authors should do more by comparing it with a non-linear competitor. I question whether the proposed model would be better than the non-linear models mentioned in the introduction.
Author Response
Thank you for your comments and suggestions.
We add a note in the corrected text (Lines 364, 419) that in addition to ARIMA models, other statistical models, including nonlinear models, were used. In addition, the SSA+LSTM model was added for comparison (Lines 382,432, Table 1), the application of which was studied in the articles described in the introduction. However, it showed relatively poor results, which is due to the short dataset and some peculiarities of the analyzed data. This does not necessarily mean that the proposed model is generally better than the nonlinear models mentioned in the introduction. However, for data similar to those studied, the proposed model gives better results.
Also, the list of references is supplemented with numbers 23 and 24 (Lines 623- 626), which are used in the "Discussion" section of the corrected text of the article.

Reviewer 3 Report
Accept in present form.
Author Response

(The authors gave the same response as above.)

Round 3
Reviewer 2 Report
I have no futher comments.